# ⚒FixingGS: Enhancing 3D Gaussian Splatting via Training-Free Score Distillation

## Abstract

Recently, 3D Gaussian Splatting (3DGS) has demonstrated remarkable success in 3D reconstruction and novel view synthesis. However, reconstructing 3D scenes from sparse viewpoints remains highly challenging due to insufficient visual information, which results in noticeable artifacts persisting across the 3D representation. To address this limitation, recent methods have resorted to generative priors to remove artifacts and complete missing content in under-constrained areas. Despite their effectiveness, these approaches struggle to ensure multi-view consistency, resulting in blurred structures and implausible details. In this work, we propose **FixingGS**, a training-free method[1] that fully exploits the capabilities of the existing diffusion model for sparse-view 3DGS reconstruction enhancement. At the core of FixingGS is our distillation approach, which delivers more accurate and cross-view coherent diffusion priors, thereby enabling effective artifact removal and inpainting. In addition, we propose an adaptive progressive enhancement scheme that further refines reconstructions in under-constrained regions. Extensive experiments demonstrate that FixingGS surpasses existing state-of-the-art methods with superior visual quality and reconstruction performance. Our code will be released publicly.

## 1 Introduction

3D reconstruction and novel view synthesis (NVS) are fundamental problems in computer vision and computer graphics, with a broad range of applications, e.g., VR/AR (Jiang et al. (2024)), autonomous driving (Zhou et al. (2024); Khan et al. (2025)), robotics (Lu et al. (2024); Zheng et al. (2024)), etc. Among recent advances, 3D Gaussian Splatting (3DGS) (Kerbl et al. (2023)) has demonstrated remarkable performance in both reconstruction quality and rendering efficiency. Despite its effectiveness, the requirement of dense support views and carefully curated captures hinders its practical applications. When constrained to sparse observations, 3DGS suffers from severe performance degradation, manifesting as noticeable artifacts and incomplete reconstructions, particularly in under-observed regions. This phenomenon arises because, under sparse input conditions, 3DGS tends to overfit the limited views and simulate view-dependent effects by introducing artifacts.

To address this limitation, previous works have introduced various forms of regularization strategies during the optimization of 3DGS (Zhu et al. (2024); Li et al. (2024); Turkulainen et al. (2025); Zhang et al. (2024)), yet these approaches remain sensitive to noise and often deliver only limited gains. In parallel, another line of research resorts to large generative models. In particular, diffusion models (DMs), which are trained on internet-scale data and have shown the remarkable capacity to generate diverse and photorealistic images, have also gained significant attention in 3D reconstruction and novel view synthesis enhancement. For instance, 3DGS-Enhancer (Liu et al. (2024)) and GenFusion (Wu et al. (2025b)) incorporate fine-tuned video diffusion models to fix the artifact-prone renderings and distill back to the 3D representation. Difix3D+ (Wu et al. (2025a)) also follows a similar process, but fine-tunes a single-step diffusion model for efficiency and further improves rendering quality by an additional post-process diffusion inference. Despite notable improvements, these approaches still

---

[1] "Training-free" in this paper means that our method does not require any additional training or fine-tuning of the diffusion model.

Figure 1: **Schematic diagram of the difference between previous methods and ours. Left:** Previous approaches update diffusion priors as pseudo ground truth of extra views at each fix rounds, keeping them unchanged in between. Until the next fix, the previous priors still act as guidance to the ongoing optimization, leading to confused supervision. **Right:** In contrast, our method dynamically distills diffusion priors throughout the optimization process, yielding more reliable guidance and significantly improved results.

face challenges in maintaining cross-view consistency, frequently resulting in blurred structures or noisy reconstructions.

Existing 3DGS enhancement methods with diffusion models typically rely on training their powerful and task-specific diffusion models with hand-crafted and carefully curated datasets, a process that is both labor-intensive and time-consuming. Moreover, they fail to fully exploit the potential of their pre-trained diffusion models. In practice, they all follow a similar protocol that updates diffusion priors at regular intervals. Since these priors are generated from previously rendered images that may suffer from severe artifacts and missing content, they can inadvertently introduce misleading supervision signals into the ongoing optimization process, thereby hindering high-fidelity reconstruction. For detailed analysis, please refer to Section 3.2.

In this work, we introduce **FixingGS**, a novel framework tailored for improving 3DGS representations under sparse-view settings. Unlike previous approaches that update diffusion priors only at fixed intervals, which may lead to misguidance, the core of FixingGS is a training-free distillation mechanism that continuously leverages the effective and timely priors from a pre-trained diffusion model, as illustrated in Figure 1. This enforces consistency across viewpoints of diffusion guidance, thereby facilitating high-quality novel view synthesis. Moreover, we empirically observe that diffusion priors become unreliable when viewpoints deviate significantly from the observed set, often hallucinating rendering content and thus producing spurious reconstructions. To mitigate this issue, we propose an adaptive progressive enhancement strategy around unreliable viewpoints. By leveraging multiple reference views, this dynamic approach strengthens supervision in under-constrained regions and further boosts reconstruction quality. Experimental results demonstrate that the proposed FixingGS achieves superior reconstruction performance, yielding cleaner and sharper rendering results.

The contributions of this paper are summarized as follows:

- A training-free distillation scheme is proposed to fully leverage the existing diffusion model and address the cross-view inconsistency issue.

- An adaptive progressive enhancement is developed, strengthening supervision around unreliable viewpoints with multiple references to improve reconstruction quality.

- Extensive experiments on multiple benchmarks demonstrate superior quantitative and qualitative performance over state-of-the-art approaches.

## 2    RELATED WORKS

**Priors for Novel View Synthesis.**    Neural Radiance Fields (NeRFs) (Mildenhall et al. (2020)) and 3D Gaussian Splatting (3DGS) (Kerbl et al. (2023)) have revolutionized the reconstruction and novel view synthesis (NVS). However, they rely on strong assumptions about the capture setup, typically requiring perfect data like dense coverage and carefully controlled conditions, which largely prohibit its practical applicability. Achieving photorealistic rendering becomes challenging from sparse and extreme novel viewpoints, with severe artifacts and missing regions in under-observed areas. Numerous works have attempted to address this issue by incorporating additional priors and regularizations into the NeRF or 3DGS optimization, including depth supervision (Deng et al. (2022); Wang et al. (2023); Zhu et al. (2024); Wang et al. (2023); Li et al. (2024); Chung et al. (2024)), normal supervision (Yu et al. (2022); Yang et al. (2023); Turkulainen et al. (2025)), smoothness constraints (Niemeyer et al. (2022); Yang et al. (2023); Zhang et al. (2024)), random dropout strategy (Park et al. (2025); Xu et al. (2025)), etc, to enhance novel view synthesis. While these methods provide incremental improvements, their effectiveness is often scene-dependent and they remain sensitive to noise, which hinders broader applicability.

**Generative Priors for Novel View Synthesis.**    Recently, generative models (Rombach et al. (2022); Sauer et al. (2024)) have made remarkable progress in generating photorealistic content. Building on this progress, a growing body of work (Weber et al. (2024); Wu et al. (2024); Paliwal et al. (2025); Liu et al. (2024); Wu et al. (2025b;a); Yin et al. (2025); Wei et al. (2025)) leverages generative priors to repair degraded regions and inpaint implausible content, thereby improving novel view synthesis. To improve temporal coherence, several works resort to video diffusion models. 3DGS-Enhancer (Liu et al. (2024)) is the pioneering work that trains a video diffusion model on a large-scale dataset, repairs extra views, and distills to the low-quality 3DGS representation. GenFusion (Wu et al. (2025b)) constructs an artifact-prone RGB-D video dataset via a masking strategy and fine-tunes a video diffusion on it for improved outpainting performance. Concurrently with our work, GSFixer (Yin et al. (2025)) continues this line of research, training a powerful video diffusion model that jointly leverages 2D semantic cues and 3D geometric features. Another representative approach is Difix3D+ (Wu et al. (2025a)), which consists of three stages: (a) training a single-step diffusion model, Difix, on hand-crafted artifact-clean image pairs; (b) distilling diffusion priors into the optimization every 2k steps, referred to as Difix3D; (c) applying additional inference-time refinement by Difix, dubbed Difix3D+. In this paper, we take a different perspective. Instead of investing in the training of more powerful diffusion models, we investigate how to fully exploit the existing diffusion model (i.e., Difix) to enhance sparse-view 3DGS reconstruction.

## 3    METHOD

Our goal is to enhance 3D Gaussian Splatting from sparse inputs. We first present the necessary preliminaries (Section 3.1), followed by an analysis of shared problems on existing 3DGS enhancement approaches with diffusion models (Section 3.2). We then detail the training-free 3DGS enhancement via score distillation (Section 3.3). Finally, we introduce an adaptive progressive enhancement (APE) that further improves the representation quality (Section 3.4).

### 3.1    PRELIMINARY

**3D Gaussian Splatting (3DGS)** (Kerbl et al. (2023)) represents a scene as a collection of explicit 3D Gaussian spheres, enabling high-quality 3D reconstruction and efficient novel view synthesis. Each Gaussian sphere $\{\mathcal{G}_i\}$ is parameterized by its position $\boldsymbol{\mu}_i \in \mathbb{R}^3$, rotation $\boldsymbol{r}_i \in \mathbb{R}^4$, scale $\boldsymbol{s}_i \in \mathbb{R}^3$, opacity $\eta_i \in \mathbb{R}$ and its view-dependent color $\boldsymbol{c}_i \in \mathbb{R}^3$ represented by sphere harmonics (SH). Each Gaussian sphere is formulated by a Gaussian function:

$$\mathcal{G}_i(x|\boldsymbol{\mu}_i, \boldsymbol{\Sigma}_i) = e^{-\frac{1}{2}(x-\boldsymbol{\mu}_i)^T \boldsymbol{\Sigma}_i^{-1}(x-\boldsymbol{\mu}_i)}, \tag{1}$$

where $\boldsymbol{\Sigma}_i$ is the corresponding 3D covariance matrix and can be decomposed into $\boldsymbol{\Sigma}_i = \boldsymbol{R}_i \boldsymbol{S}_i \boldsymbol{S}_i^T \boldsymbol{R}_i^T$, $\boldsymbol{S}_i$ and $\boldsymbol{R}_i$ denote the scaling matrix and rotation matrix correponding to $\boldsymbol{s}_i$ and $\boldsymbol{r}_i$ respectively. Novel view can be rendered by fast $\alpha$-blending rendering, defined as:

$$C = \sum_{i \in M} c_i \alpha_i \prod_{j=1}^{i-1} (1 - \alpha_j), \tag{2}$$

where $C$ denotes the final pixel color, $\alpha_i$ is calculated by evaluating $\mathscr{G}_i(x)$ multiplied with $\eta_i$, $M$ is the number of Gaussian spheres that overlap with the pixel on the 3D camera planes.

**Diffusion models (DMs)** (Ho et al. (2020); Sohl-Dickstein et al. (2015); Song et al. (2020)) are a series of generative models that generate data by iteratively denoising from pure Gaussian noise by learning a distribution of data $p_\theta(x)$. DMs consist of two stages: the forward diffusion stage and the reverse denoising stage. During the forward diffusion stage, DMs progressively add Gaussian noise $\varepsilon \sim \mathcal{N}(0, \mathbf{I})$ to the clean data $x_0$ to obtain the diffused version $x_t = \alpha_t x_0 + \sigma_t \varepsilon$, where $\alpha_t$ and $\sigma_t$ represent the noise schedule coefficients at timestep $t$. The reverse denoising process learns the distribution $p_\theta(x)$ with a noise predictor $\varepsilon_\theta$ to recover the original data by removing noise. The noise predictor is trained to minimize the denoising objective as:

$$\min_\theta \mathbb{E}_{t \sim \mathcal{U}(0,1), \varepsilon \sim \mathcal{N}(\mathbf{0}, \mathbf{I})} [||\varepsilon_\theta(x_t; c, t) - \varepsilon||_2^2], \tag{3}$$

where $c$ denotes optional conditioning information (e.g., text prompts or image content).

## 3.2 ANALYSIS OF PREVIOUS 3DGS ENHANCEMENT METHODS USING DIFFUSION MODELS

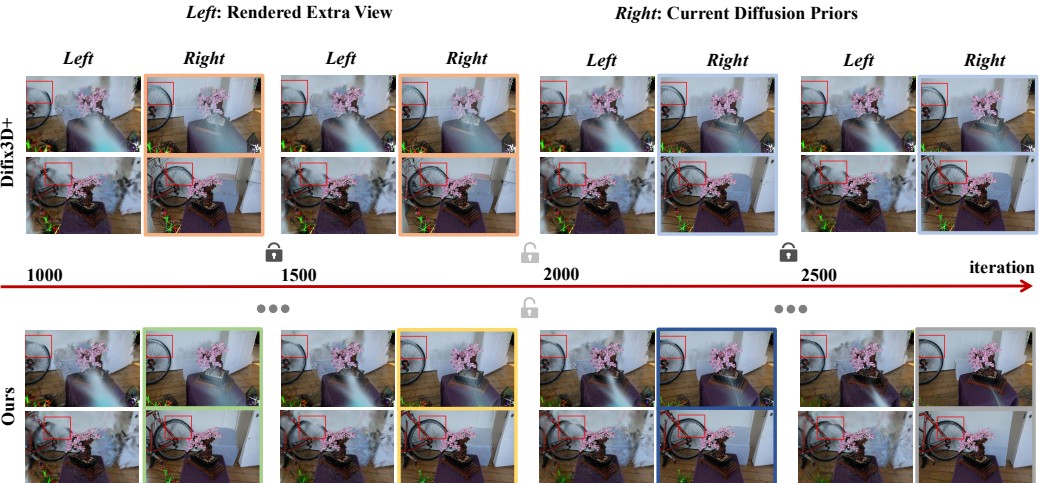

Figure 2: **Illustration of the main difference between Difix3D+ (Wu et al. (2025a)) and our proposed method.** We compare two adjacent rendered extra views and corresponding diffusion priors at iterations 1000, 1500, 2000, 2500, for instance. We color the borders of diffusion priors, with the same color indicating the same diffusion priors. The same 3D region is highlighted with red bounding boxes. **Top:** Difix3D+ updates diffusion priors only every 2000 steps, leaving them unchanged in between, which results in misleading guidance. This progressive approach results in notable artifacts and multi-view inconsistency (e.g., the bicycle wheel and bottom-left artifacts). **Bottom:** Our approach instead continuously distills diffusion priors throughout optimization, fully exploiting the diffusion model for accurate guidance, which yields improved cross-view consistency and cleaner renderings.

Recently, leveraging diffusion models to enhance 3DGS quality from limited inputs has gained increasing attention. Given an initial low-quality 3D representation, representative works such as 3DGS-Enhancer (Liu et al. (2024)), Difix3D+ (Wu et al. (2025a)), and GenFusion (Wu et al. (2025b)) generally follow a similar pipeline:

- Train a novel diffusion model tailored for enhancing artifact-prone rendered images on carefully curated datasets.

- At fixed intervals of $M$ iterations (where $M$ varies across different methods), repair all rendered images from extra viewpoints using the trained diffusion model. Then add the repaired views to the training set.
- Optimize the 3DGS representation with the current training set and repeat the process until convergence.

While these methods yield notable improvements, they fail to fully exploit the capacity of diffusion models. Under sparse-view conditions, regions with limited observations often suffer from severe degradation. Although recent diffusion models are effective at enhancing artifact-prone images, recovering accurate cross-view consistent content in heavily degraded regions remains a challenge. In existing methods, diffusion priors are updated only every $M$ iterations, remaining unchanged in between. These repaired images, derived from low-quality renderings from the previous steps, become lagged and unreliable priors in subsequent $M$ steps optimization, often misguiding the process toward multi-view inconsistency and ambiguous results, as illustrated in detail in Figure 2. Following this protocol, even if we invest significant time and effort into training more powerful diffusion models, we still cannot overcome this limitation.

Analyzing this limitation of previous works motivates us to rethink how pre-trained diffusion models should be utilized. To this end, we introduce a distillation strategy that continuously incorporates diffusion priors throughout the optimization process, without requiring additional diffusion model training (Section 3.3).

### 3.3 TRAINING-FREE SCORE DISTILLATION FOR 3DGS ENHANCEMENT

We begin by revisiting score distillation sampling (SDS), which underpins our design. Originally introduced in DreamFusion (Poole et al. (2023)), SDS distills guidance from a 2D pre-trained diffusion model to optimize a 3D representation parameterized by $\theta$. We denote the diffusion model as $\varepsilon_\phi(x_t, t, y)$ with extra condition $y$ and timestep $t$. Given a camera pose $c_i$, an image is rendered from 3DGS by a differentiable rendering function $g(\theta, c_i)$. In the original SDS setting, the rendered image $x = g(\theta, c_i)$ is used to optimize $\theta$ through the following gradient:

$$\nabla_\theta \mathscr{L}_{SDS} = \mathbb{E}_{t,\varepsilon,c}[\omega(t)(\varepsilon_\phi(x_t, t, y) - \varepsilon)\frac{\partial g(\theta, c_i)}{\partial \theta}], \tag{4}$$

where $x_t = \alpha_t g(\theta, c_i) + \sigma_t \varepsilon$, and $\omega(t)$ is a weighting function.

Thanks to the full differentiability of 3DGS, we can directly optimize $\theta$ via score distillation. In this work, we primarily adopt Difix (Wu et al. (2025a)) as the pre-trained diffusion model for distillation. Difix treats artifacts in renderings as Gaussian noise in the denoising process of the original diffusion model, effectively serving as an image enhancer. In this case, we follow (Zhu (2023); Zhu et al. (2023)) and employ an image residual formulation instead of the original noise residual formulation. For extra views, we formulate the distillation loss as:

$$\mathscr{L}_{distillation} = \left\|\omega(t_0)\big(g(\theta, c) - \mathscr{D}_\phi(g(\theta, c); t_0, y)\big)\right\|_2^2, \tag{5}$$

where $\mathscr{D}_\phi(g(\theta, c); t_0, y)$ denotes the recovered image from Difix, and $y$ represents the clean reference image. We set $t_0 = 199$, following the official configuration of Difix (Wu et al. (2025a)), and assign $\omega(t_0) = 0.5$.

For training views, we maintain the photorealistic loss function as the original implementation of 3DGS, defined as:

$$\mathscr{L}_{photo} = \lambda_{l1}\mathscr{L}_{l1} + \lambda_{SSIM}\mathscr{L}_{SSIM}, \tag{6}$$

where $\lambda_{l1}$ and $\lambda_{SSIM}$ are set to 0.2 and 0.8, respectively.

### 3.4 ADAPTIVE PROGRESSIVE ENHANCEMENT AROUND UNRELIABLE VIEWS

As a rendering enhancer, the fixing ability of the diffusion model is inherently limited by the quality of renderings. When the desired novel views lie far from or are weakly constrained by the input

---

**Algorithm 1:** Adaptive Progressive Enhancement (APE)

---

**Input:** Initial 3DGS parameters $\theta$, Differentiable rendering function $g(\theta, c)$ given a camera pose $c$, Pre-trained diffusion model (DM) $\mathscr{D}_\phi$, Number of iterations per enhancement $N_{iter}$, Number of reference $M$, Unreliable threshold $\eta$, Training view poses $C_{train}$, Extra view poses $C_{extra}$.

1 **Definition:** Pose distance calculator: $dist(\cdot)$, PSNR calculator: $psnr(\cdot)$, Pose shifting: $shift(\cdot)$
2 **while** *not converged* **do**
3    **for** $i = 1$ *to* $N_{iter}$ **do**
4       Optimize $\theta$ using the current training set.
5    **for** *each* $c \in C_{extra}$ **do**
6       $I_{extra} \leftarrow g(\theta, c)$ ;                 /* Render the extra view */
7       $[I_{ref_1}, ..., I_{ref_M}] \leftarrow dist(C_{train}, c)[: M]$ ; /* Find $M$ nearest reference views */
8       $I_{fix} \leftarrow \mathscr{D}_\phi(I_{extra}; I_{ref_1})$ ;    /* Obtain the fixed extra image via DM */
9       **if** $psnr(I_{fix}, I_{extra}) < \eta$ **then**
10          **for** *each* $i_{ref} \in [I_{ref_1}, ..., I_{ref_M}]$ **do**
11             $c_{shift} \leftarrow shift(c_{ref}, c, i)$ ;   /* $c_{ref}$ is the related pose of $i_{ref}$ */
12             $I_{shift} \leftarrow g(\theta, c_{shift})$ ;         /* Render the shifted view */
13             $I_{novel} \leftarrow \mathscr{D}_\phi(I_{shift}; i_{ref})$ ; /* Obtain fixed novel view via DM */
14             Add $I_{novel}$ to the training set.

---

observations, their renderings often suffer from severe artifacts and missing regions. In such cases, the diffusion model struggles to recover reliable high-fidelity details and instead tends to hallucinate, thereby providing unreliable guidance for optimization.

To address this challenge, we propose an adaptive progressive enhancement (APE) strategy to further strengthen supervision around unreliable views. As outlined in Algorithm 1, when a viewpoint is identified as unreliable, APE leverages multiple training views as stronger references and applies pose perturbations toward the target viewpoint. This adaptive design progressively improves the quality of novel view renderings. By jointly exploiting multiple references and an adaptive selection mechanism, FixingGS with APE significantly outperforms Difix3D, as shown in Section 4.3. More details are provided in the Appendix.

## 4 EXPERIMENTS

We first describe the experimental setup used to evaluate FixingGS (Section 4.1). We then present quantitative and qualitative comparisons with state-of-the-art 3DGS reconstruction enhancement methods (Section 4.2). Finally, we conduct ablation studies to analyze the contribution of each component (Section 4.3).

### 4.1 EXPERIMENTAL SETUP

**Evaluation Dataset.** We evaluate FixingGS on two challenging real-world datasets: 10 scenes from DL3DV-10K (Ling et al. (2024)) and 9 scenes from Mip-NeRF 360 (Barron et al. (2022)). Both datasets cover indoor and outdoor scenarios. For DL3DV-10K, we randomly select 10 scenes and uniformly sample training views along the camera trajectory, while test views are chosen every 8 views from the remaining held-out set. For Mip-NeRF360, we adopt the same data partitioning protocol as ReconFusion (Wu et al. (2024)).

**Metrics.** For metrics, we calculate commonly-used PSNR, SSIM (Wang et al. (2004)), as well as LPIPS (Johnson et al. (2016)) on novel views to measure 3D reconstruction quality and fidelity.

**Baselines.** We compare our FixingGS against its backbone method, 3DGS (Liu et al. (2024)), as well as several recent state-of-the-art approaches for 3DGS reconstruction enhancement, including FSGS (Zhu et al. (2024)), GenFusion (Wu et al. (2025b)), and Difix3D+ (Wu et al. (2025a)). For

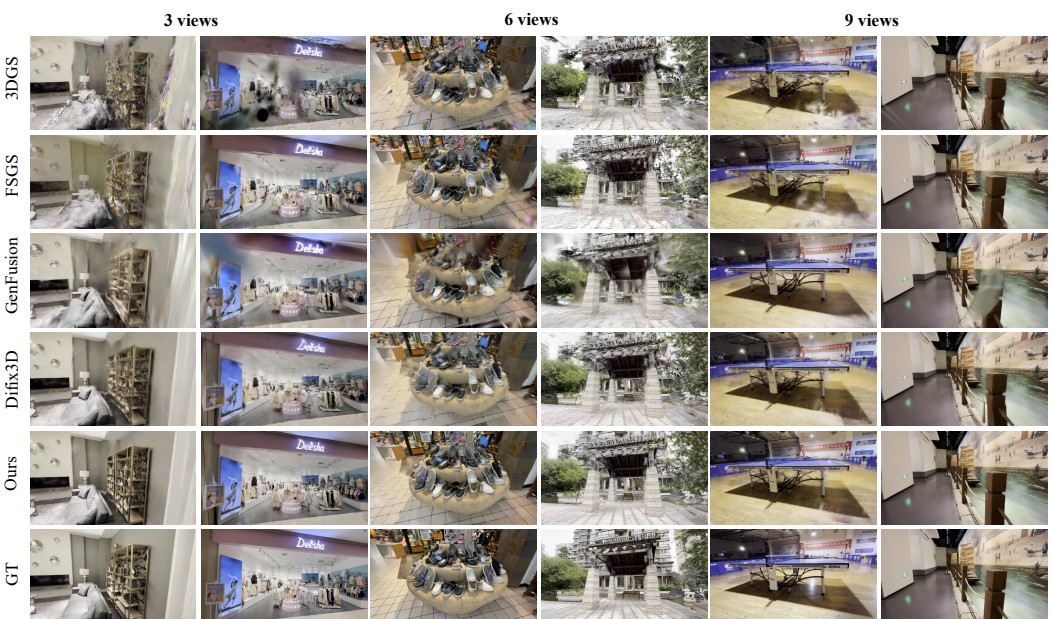

Figure 3: **Qualitative Comparison on the DL3DV-10K dataset (Ling et al. (2024)).**

| | PSNR ↑ | | | | SSIM ↑ | | | | LPIPS ↓ | | | |
|---|---|---|---|---|---|---|---|---|---|---|---|---|
| | 3-view | 6-view | 9-view | Avg. | 3-view | 6-view | 9-view | Avg. | 3-view | 6-view | 9-view | Avg. |
| 3DGS (Kerbl et al. (2023)) | 16.40 | 19.70 | 21.28 | 19.13 | 0.588 | 0.709 | 0.737 | 0.678 | 0.498 | 0.321 | 0.252 | 0.357 |
| FSGS (Zhu et al. (2024)) | 16.98 | 20.47 | 23.01 | 20.15 | 0.645 | 0.740 | 0.802 | 0.729 | 0.437 | 0.322 | 0.258 | 0.339 |
| GenFusion (Wu et al. (2025b)) | 15.97 | 20.49 | 23.02 | 19.83 | 0.615 | 0.750 | 0.814 | 0.726 | 0.438 | 0.311 | 0.248 | 0.332 |
| Difix3D (Wu et al. (2025a)) | 16.86 | 20.59 | 23.13 | 20.19 | 0.609 | 0.731 | 0.799 | 0.713 | 0.417 | 0.270 | 0.197 | 0.295 |
| Difix3D+ (Wu et al. (2025a)) | 16.45 | 20.03 | 22.54 | 19.67 | 0.583 | 0.709 | 0.778 | 0.690 | 0.393 | 0.287 | 0.230 | 0.303 |
| FixingGS (Ours) | 17.67 | 21.28 | 23.73 | 20.89 | 0.648 | 0.760 | 0.824 | 0.744 | 0.396 | 0.239 | 0.174 | 0.270 |

Table 1: **Quantitative comparison on the DL3DV-10K dataset (Ling et al. (2024)).** We compare the rendering quality with baselines given 3, 6, and 9 views. Each column is colored as: best and second best .

experiments on the Mip-NeRF360 dataset, we further include representative NeRF-based baselines (Barron et al. (2023); Yang et al. (2023); Somraj et al. (2023); Sargent et al. (2024); Wu et al. (2024)) for comprehensive comparison.

The official implementation of Difix3D+ incorporates an additional diffusion inference step as a rendering enhancer. We follow the official implementation using its open-sourced code to evaluate our experimental setup. However, under our conditions, we observe a noticeable drop in performance, contrary to the claims reported in the paper. Please refer to the Appendix for explanations. Meanwhile, it is important to note that our method does not rely on any inference-time enhancement. For a fair comparison, we also report results from Difix3D (i.e., Difix3D+ without the additional inference step).

**Implementation Details.**    In our framework, we use Difix (Wu et al. (2025a)) as the pre-trained diffusion model for both distillation and adaptive progressive enhancement (APE). FixingGS is trained for 6,000 steps in all experiments. Our empirical findings indicate that diffusion priors tend to stabilize in the later stages of optimization, with variations progressively diminishing. To enhance efficiency, we freeze the priors once they converge without noticeable changes. All results are obtained using a single NVIDIA RTX 3090 GPU. Our implementation is based on PyTorch. Discussions on the associated training-time trade-offs are provided in the Appendix.

## 4.2    COMPARISON WITH STATE-OF-THE-ARTS

Qualitative and quantitative comparisons on the DL3DV-10K dataset are reported in Figure 3 and Table 1, while results on the MipNeRF360 dataset are shown in Figure 4 and Table 2. Numerical results (Table 1 and Table 2) on both datasets shows that our method consistently outperforms all

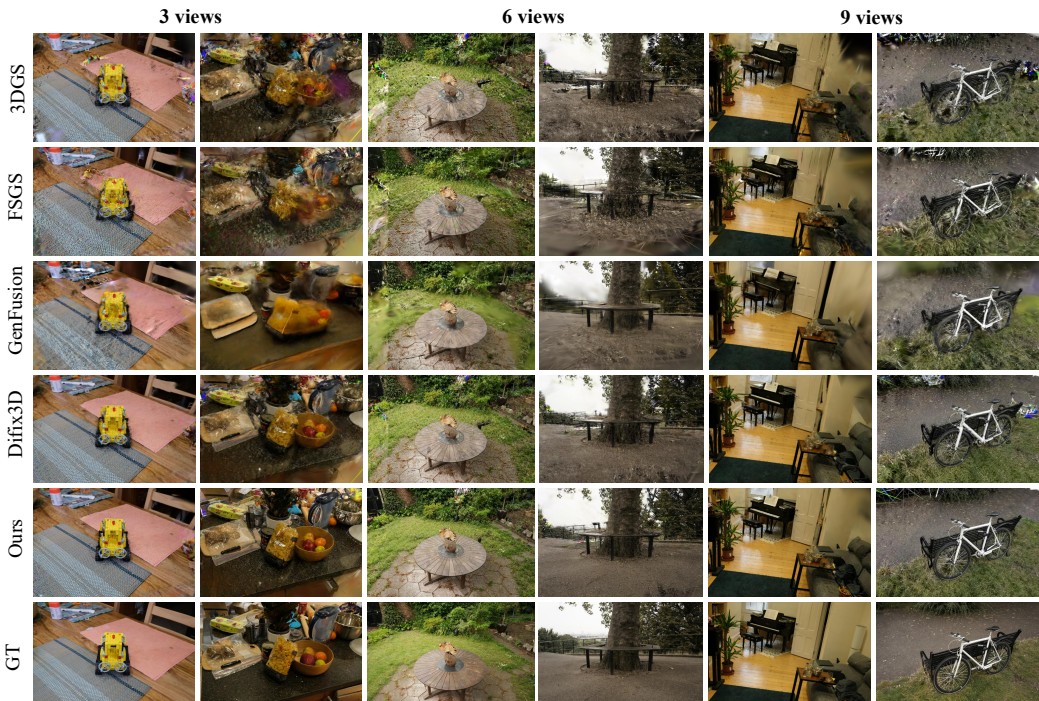

Figure 4: **Visual Comparisons on the Mip-NeRF 360 dataset (Barron et al. (2022)).**

| | PSNR ↑ | | | | SSIM ↑ | | | | LPIPS ↓ | | | |
|---|---|---|---|---|---|---|---|---|---|---|---|---|
| | 3-view | 6-view | 9-view | Avg. | 3-view | 6-view | 9-view | Avg. | 3-view | 6-view | 9-view | Avg. |
| ZipNeRF[†] (Barron et al. (2023)) | 12.77 | 13.61 | 14.30 | 13.56 | 0.271 | 0.284 | 0.312 | 0.289 | 0.705 | 0.663 | 0.633 | 0.667 |
| FreeNeRF[†] (Yang et al. (2023)) | 12.87 | 13.35 | 14.59 | 13.60 | 0.260 | 0.283 | 0.319 | 0.287 | 0.715 | 0.717 | 0.695 | 0.709 |
| SimpleNeRF[†] (Somraj et al. (2023)) | 13.27 | 13.67 | 15.15 | 14.03 | 0.283 | 0.312 | 0.354 | 0.316 | 0.741 | 0.721 | 0.676 | 0.713 |
| ZeroNVS[†] (Sargent et al. (2024)) | 14.44 | 15.51 | 15.99 | 15.31 | 0.316 | 0.337 | 0.350 | 0.334 | 0.680 | 0.663 | 0.655 | 0.666 |
| ReconFusion[†] (Wu et al. (2024)) | 15.50 | 16.93 | 18.19 | 16.87 | 0.358 | 0.401 | 0.432 | 0.397 | 0.585 | 0.544 | 0.511 | 0.547 |
| 3DGS[†] (Kerbl et al. (2023)) | 13.06 | 14.96 | 16.79 | 14.94 | 0.251 | 0.355 | 0.447 | 0.351 | 0.576 | 0.505 | 0.446 | 0.509 |
| FSGS[†] (Zhu et al. (2024)) | 14.17 | 16.12 | 17.94 | 16.08 | 0.318 | 0.415 | 0.492 | 0.408 | 0.578 | 0.517 | 0.468 | 0.521 |
| GenFusion[†] (Wu et al. (2025b)) | 15.29 | 17.16 | 18.36 | 16.93 | 0.369 | 0.447 | 0.496 | 0.437 | 0.585 | 0.500 | 0.465 | 0.517 |
| Difix3D[‡] (Wu et al. (2025a)) | 15.05 | 17.26 | 18.36 | 16.89 | 0.357 | 0.449 | 0.510 | 0.439 | 0.479 | 0.371 | 0.320 | 0.390 |
| Difix3D+[‡] (Wu et al. (2025a)) | 14.72 | 16.85 | 17.81 | 16.46 | 0.315 | 0.406 | 0.455 | 0.392 | 0.490 | 0.422 | 0.386 | 0.433 |
| FixingGS (Ours) | 15.78 | 17.72 | 18.87 | 17.46 | 0.376 | 0.464 | 0.523 | 0.454 | 0.483 | 0.383 | 0.303 | 0.390 |

Table 2: **Quantitative comparison on the Mip-NeRF 360 dataset (Barron et al. (2022)).** We compare the rendering quality with baselines given 3, 6, and 9 views. † denotes results reproduced by ReconFusion and GenFusion; while ‡ denote results reproduced by us on their official implemantation.

baselines across almost all evaluation metrics (e.g., at least 0.7dB and 0.5dB PSNR improvement over state-of-the-art methods in DL3DV-10K and Mip-NeRF 360 datasets, respectively), indicating that our method reconstructs the highest-quality and most faithful scenes.

Visual comparisons (Figure 3 and Figure 4) more clearly highlight the strengths of our method. Specifically, 3DGS (Kerbl et al. (2023)) and FSGS (Zhu et al. (2024)) exhibit severe degradation, with noticeable artifacts persisting in the reconstructed scenes. GenFusion (Wu et al. (2025b)) mitigates artifacts by leveraging a fine-tuned video diffusion model, but frequently produces overly smoothed geometry. Difix3D (Wu et al. (2025a)) achieves improved artifact removal through their powerful diffusion model as priors, yet struggles to recover fine structural details and introduces ambiguous results. In contrast, our approach yields sharper reconstructions with significantly fewer artifacts and more high-frequency structures, highlighting its advantage in preserving high-fidelity structures from limited viewpoint inputs. For fair comparison, we additionally show visual results of Difix3D+ (i.e., Difix3D with an extra diffusion enhancement) and our method with the same procedure in the Appendix. In summary, both quantitative and qualitative comparisons with state-of-the-art baselines demonstrate the strong potential of our approach to substantially improve the quality and fidelity of novel view synthesis.

|  | DL3DV-10K | | | Mip-NeRF 360 | | |
|---|---|---|---|---|---|---|
|  | PSNR ↑ | SSIM ↑ | LPIPS ↓ | PSNR ↑ | SSIM ↑ | LPIPS ↓ |
| *full model* | **20.89** | **0.744** | **0.270** | **17.46** | **0.454** | **0.390** |
| *w/o distillation* | 20.54 | 0.734 | 0.303 | 17.03 | 0.446 | 0.413 |
| *w/o APE* | 20.74 | 0.735 | 0.279 | 17.25 | 0.446 | 0.408 |
| Difix3D | 20.19 | 0.713 | 0.295 | 16.89 | 0.439 | 0.390 |

Table 3: **Ablation study of FixingGS on both datasets.** The quantitative results are averaged across 3, 6, and 9 views.

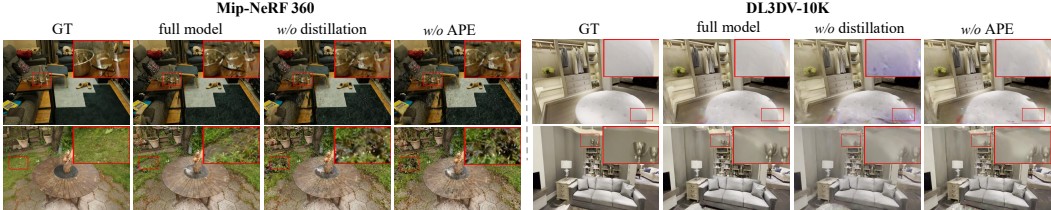

Figure 5: **Qualitative ablation results of FixingGS on both datasets.** We highlight the most prominent differences in red bounding boxes.

## 4.3 ABLATION STUDY

We conduct the ablation experiments with entire scenes and sparse-view conditions to validate the effectiveness of each component. We present numerical results in Table 3 and visual performance in Figure 5. We compare our full model with two alternatives: a variant without our distillation approach (dubbed *w/o* distillation), and a variant without the adaptive progressive enhancement (dubbed *w/o* APE). In addition, we also compare Difix3D to further demonstrate the effectiveness of APE. Please refer to the Appendix to see the difference between Difix3D and our proposed APE.

**Effectiveness of our distillation approach.** To analyze the impact of our distillation method, which serves as the core contribution of this paper, we ablate this design. Without the distillation, FixingGS shows a notable decline in all metrics. Visual comparisons further demonstrate its effectiveness. Without the distillation strategy, our method struggles to inpaint the missing regions and fails to eliminate artifacts in representations. Incorporating this contribution can fully benefit robust priors from diffusion models, yielding promising artifact-removal and inpainting performance. These results highlight the necessity and effectiveness of our distillation approach.

**Effectiveness of APE.** To assess the effectiveness of the proposed enhancement on unreliable viewpoints, we perform an ablation by disabling this component. The full FixingGS consistently outperforms the ablated variant across all evaluation metrics. For further validation, we also compare with Difix3D. By further adaptively targeting unreliable viewpoints and leveraging multiple references, APE achieves substantial improvements. Visual comparisons in Figure 5 further highlight the benefit: without the enhancement strategy, the model struggles to recover fine details and produces blurrier renderings, whereas our full method yields cleaner results with fewer artifacts.

## 5 CONCLUSION

We present FixingGS, a novel framework for enhancing sparse-view 3D reconstruction. Unlike previous approaches that rely on training increasingly powerful diffusion models, our key insight is to fully exploit the capabilities of existing pre-trained diffusion models. At the core of FixingGS is a score distillation strategy that effectively mitigates the long-standing issue of multi-view inconsistency in previous reconstruction enhancement works with diffusion priors, leading to substantial improvements in reconstruction quality. In addition, we propose an adaptive progressive enhancement around unreliable viewpoints that further refines reconstruction in under-constrained regions. We conduct extensive experiments, demonstrating our superior improvement in producing high-quality and multi-view consistent reconstructions.

## 6 ETHICS STATEMENT

This work adheres to the ICLR Code of Ethics. Our experiments use only publicly available datasets that do not contain personally identifiable or sensitive information. We provide full implementation details to support transparency and reproducibility. The proposed method, FixingGS, is intended to improve sparse-view 3D reconstruction for research and practical applications. While the method does not directly raise privacy or fairness concerns, any generative approach may be misused to create synthetic 3D content that could be applied irresponsibly. We encourage responsible use of this work and emphasize that it should not be applied in harmful contexts.

## 7 REPRODUCIBILITY STATEMENT

We provide full implementation of our method, including training and evaluation code, as supplementary material, and will release it publicly upon acceptance.

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

# APPENDIX: FIXINGGS: ENHANCING 3D GAUSSIAN SPLATTING VIA TRAINING-FREE SCORE DISTILLATION

## A  LARGE LANGUAGE MODELS (LLMs) USAGE

In preparing this paper, we use a large language model (LLM) solely as a writing assistant to help polish the clarity and readability of the text. The LLM was not involved in research ideation, experimental design, data analysis, or the development of technical contributions. All scientific content, methodology, experiments, and results presented in this paper are entirely the work of the authors. The authors take full responsibility for the contents of the paper.

## B  MORE DETAILS IN APE

Here we present details on APE. Pseudocode is provided in the main paper. In this method, we apply Difix as the pre-trained diffusion model. We set the unreliable threshold $\eta = 25$ dB; number of reference $M = 3$; number of iterations per enhancement $N_{iter} = 1000$. We define the pose distance calculator $dist(\cdot)$, PSNR calculator $psnr(\cdot)$ and pose $shifting(\cdot)$ as follow:

**Definition of Pose Distance Calculator** $dist(\cdot)$**:** Given two camera poses $P_1 = (R_1, t_1)$ and $P_2 = (R_2, t_2)$, represented as $4 \times 4$ transformation matrices with rotation $R \in SO(3)$ and translation $t \in \mathbb{R}^3$, the 6-DoF pose distance calculator $dist(\cdot, \cdot)$ is defined as

$$dist(P_1, P_2) = \alpha \|t_1 - t_2\|_2 + \beta \, d_R(R_1, R_2),$$

where $\alpha$ and $\beta$ are weighting factors for translation and rotation, respectively. Here, we set $\alpha = \beta = 0.5$. The rotation distance $d_R(R_1, R_2)$ is computed from the corresponding unit quaternions $q_1, q_2$ of $R_1, R_2$ as

$$d_R(R_1, R_2) = 2 \arccos \left( |\langle q_1, q_2 \rangle| \right),$$

with $\langle q_1, q_2 \rangle$ denoting the dot product of the two quaternions.

**Definition of PSNR Calculator** $PSNR(\cdot)$**:** Given a reference image $I \in \mathbb{R}^{H \times W \times C}$ and a reconstructed image $\hat{I} \in \mathbb{R}^{H \times W \times C}$, the peak signal-to-noise ratio (PSNR) is defined as

$$PSNR(I, \hat{I}) = 10 \cdot \log_{10} \left( \frac{MAX^2}{MSE(I, \hat{I})} \right),$$

where $MAX$ is the maximum possible pixel value (e.g., $MAX = 1$ if images are normalized), and

$$MSE(I, \hat{I}) = \frac{1}{HWC} \sum_{u=1}^{H} \sum_{v=1}^{W} \sum_{c=1}^{C} \left( I(u, v, c) - \hat{I}(u, v, c) \right)^2$$

is the mean squared error between $I$ and $\hat{I}$.

**Definition of Pose Shifting** $shift(\cdot)$**:** Given a training pose $P_{\text{train}} = (R_{\text{train}}, t_{\text{train}})$ and a extra pose $P_{\text{extra}} = (R_{\text{extra}}, t_{\text{extra}})$, the pose shifting operator $shift(P_{\text{train}}, P_{\text{test}}, \tau)$ generates an interpolated pose based on the current progress $\tau \in [0, 1]$ of the optimization process as

$$shift(P_{\text{train}}, P_{\text{extra}}, \tau) = \left( R(\tau), \, t(\tau) \right),$$

where

$$t(\tau) = (1 - \tau) t_{\text{train}} + \tau t_{\text{extra}},$$
$$R(\tau) = \text{Slerp}(R_{\text{train}}, R_{\text{extra}}; \tau),$$

with Slerp$(\cdot)$ denoting spherical linear interpolation between two rotations.

## C  DIFFERENCES BETWEEN DIFIX3D AND OUR PROPOSED APE

In the framework of Difix3D+ (Wu et al. (2025a)), a strategy termed progressive 3D updates is employed with their pre-trained diffusion model (i.e., Difix). Difix3D is the 3DGS framework that applies this strategy. Concretely, every 2k iterations, pose perturbations are applied from training

views toward the target views by a fixed distance. The artifact-prone images rendered from these perturbed novel views are then repaired using Difix (with the nearest training view as reference) and subsequently added to the training set.

However, this design exhibits several limitations. We empirically observe that when viewpoints are too distant or insufficiently constrained by other views, their renderings suffer from severe degradation. In such cases, the corresponding diffusion priors become unreliable, as the diffusion model may preserve or even hallucinate more of the degradations. Moreover, Difix3D applies pose perturbations to all target views simultaneously, which often introduces misleading guidance. In addition, this strategy relies on only a single nearest training view as reference, which proves inadequate for effective reconstruction enhancement.

To address these issues, APE introduces several improvements. (a) We adopt an adaptive scheme: instead of perturbing poses toward all target views, we selectively shift only those viewpoints identified as unreliable based on their rendering quality. (b) We incorporate multiple reference views rather than relying on a single one, providing richer information as the reference for reconstruction. (c) We further refine the shifting mechanism by making the perturbation distance adaptive to both the pose distance and the optimization iteration. The comparisons in the Ablation Study (Difix3D v.s. *w/o* distillation) also demonstrate that our APE outperforms Difix3D by a significant margin.

## D  EXPLANATIONS ON PERFORMANCE DROP OF DIFIX3D+

The official GitHub repository for Difix3D+ (Wu et al. (2025a)) is available at https://github.com/nvlabs/Difix3D . As reported in the paper, Difix3D+ includes an additional inference procedure to further enhance rendering quality, corresponding to "Difix3D+: With real-time post-rendering" in the repository. However, the authors do not provide the checkpoint file of their Difix model (i.e., model.pkl). By examining their code, we found that they use the model checkpoint available on HuggingFace (https://huggingface.co/nvidia/difix_ref ) during the training of Difix3D (i.e., the Progressive 3D update in the repository). We adopt the same HuggingFace checkpoint for the additional inference procedure to repair renderings in our experimental setup. Nevertheless, our results show a numerical drop in both datasets, contrary to the claims reported in their paper.

We analyze this unexpected phenomenon in detail. The Difix3D+ paper does not specify the exact training conditions (e.g., sparse-view or dense-view) on the DL3DV-10K dataset, making Table 2 in their paper difficult to reproduce. Our work focuses on sparse-view 3DGS reconstruction, and we adopt a sparse-view setting for fair comparison with all baselines. Under these conditions, artifacts and missing content persist more prominently in under-constrained regions. While the additional inference step in Difix3D+ can remove minor artifacts, it also amplifies ambiguous regions and over-sharpens the images, leading to larger deviations from the ground truth and poorer numerical performance. Extensive per-scene visual comparisons (Figure 7 and Figure 6) further support these observations.

## E  LIMITATIONS AND FUTURE WORKS

The performance of FixingGS still inherently depends on the effectiveness of pre-trained diffusion models used for 3D reconstruction enhancement. In this work, we validate FixingGS primarily with Difix, and exploring integration with stronger or domain-specific diffusion priors represents an exciting avenue for future research. Moreover, our distillation introduces a moderate training-time overhead compared with Difix3D+ (Wu et al. (2025a)) as reported in Appendix F. Designing more efficient distillation techniques to mitigate this cost will be an important direction moving forward.

## F  TRADE-OFF BETWEEN TRAINING EFFICIENCY AND EFFECTIVENESS

As noted in the limitations, our distillation approach incurs some training-time overhead. To quantify this, we evaluate our method and baseline approaches with diffusion priors in terms of training time and GPU memory usage. As shown in Table 4, FixingGS introduces only modest overhead while delivering substantial improvements in 3DGS reconstruction and novel view synthesis quality.

| Methods | PSNR↑ | SSIM↑ | LPIPS ↓ | Training Time↓ | Memory Usage (GiB)↓ |
|---------|-------|-------|---------|----------------|---------------------|
| Difix3D+ | 16.46 | 0.392 | 0.433 | $\sim$ 20 min | 12.14 |
| GenFusion | 16.93 | 0.437 | 0.517 | $\sim$ 28 min | 23.69 |
| Ours | 17.46 | 0.454 | 0.390 | $\sim$ 29 min | 11.95 |

Table 4: Evaluations of training efficiency (presented in Training Time and Memory Usage) and novel view synthesis quality (presented in PSNR, SSIM, and LPIPS) on the Mip-NeRF 360 dataset.

## G  EVALUATION ON MULTI-VIEW CONSISTENCY

We evaluate FixingGS using the Thresholded Symmetric Epipolar Distance (TSED) metric (Yu et al. (2023)), which measures the consistency of frame pairs within a sequence. As shown in Table 5, our method achieves higher TSED values than the baselines, indicating stronger multi-view consistency.

|  | 3DGS | Difix3D | Ours |
|--|------|---------|------|
| 3 views | 0.4286 | 0.4408 | **0.4673** |
| 6 views | 0.4286 | 0.4367 | **0.4551** |
| 9 views | 0.4347 | 0.4367 | **0.4551** |

Table 5: Multi-view consistency evaluations on the DL3DV dataset. Higher TSED values indicate better multi-view consistency performance.

## H  ADDITIONAL VISUAL COMPARISONS

Note that our proposed FixingGS does not apply the additional diffusion inference. For fair comparison, we also provide visual results of Difix3D+ and Ours+ (i.e., FixingGS with the same additional inference procedure). We present extensive per-scene visual results in Figure 6 and Figure 7.

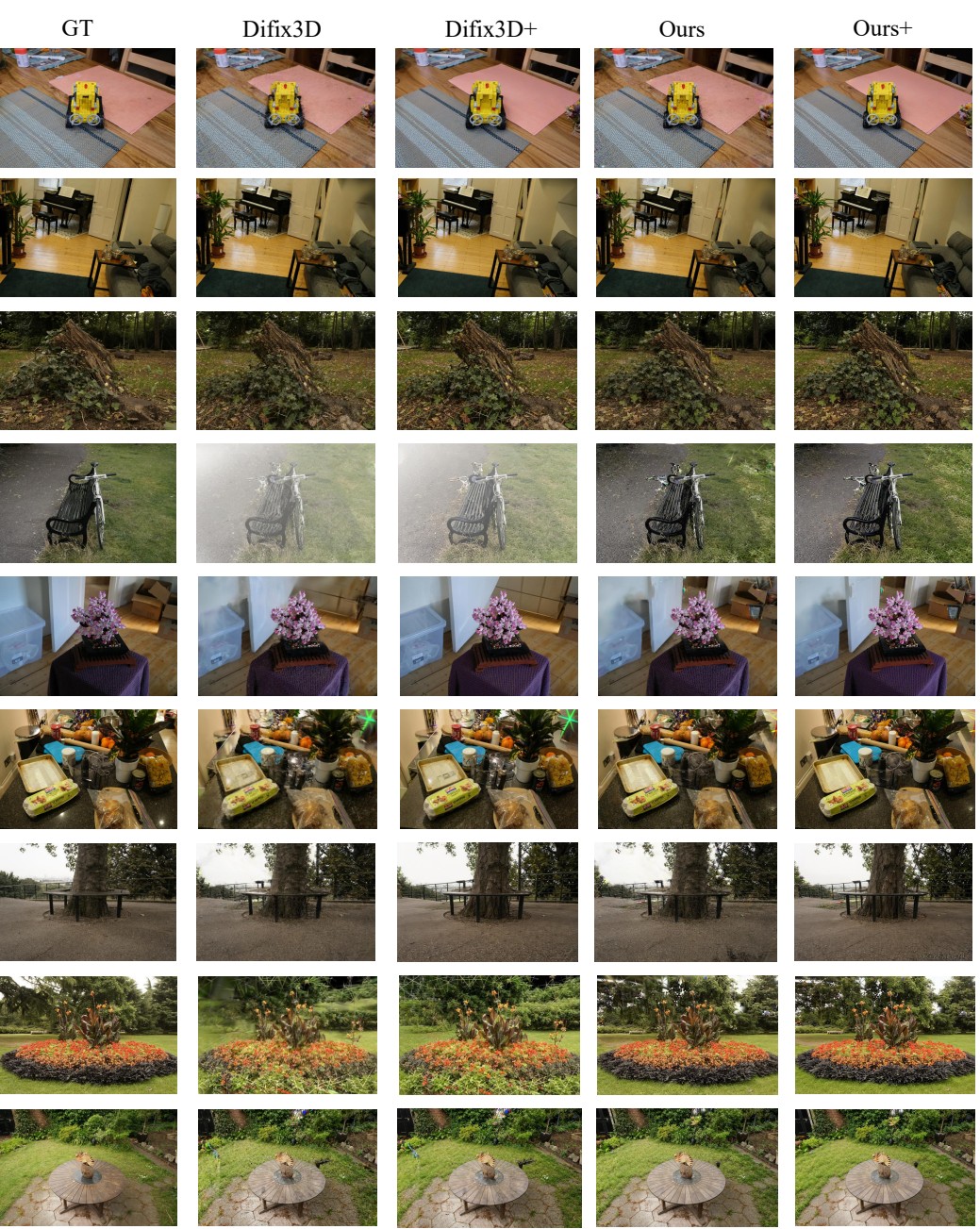

Figure 6: **More visual comparisons on the Mip-NeRF 360 dataset (Barron et al. (2022)).**

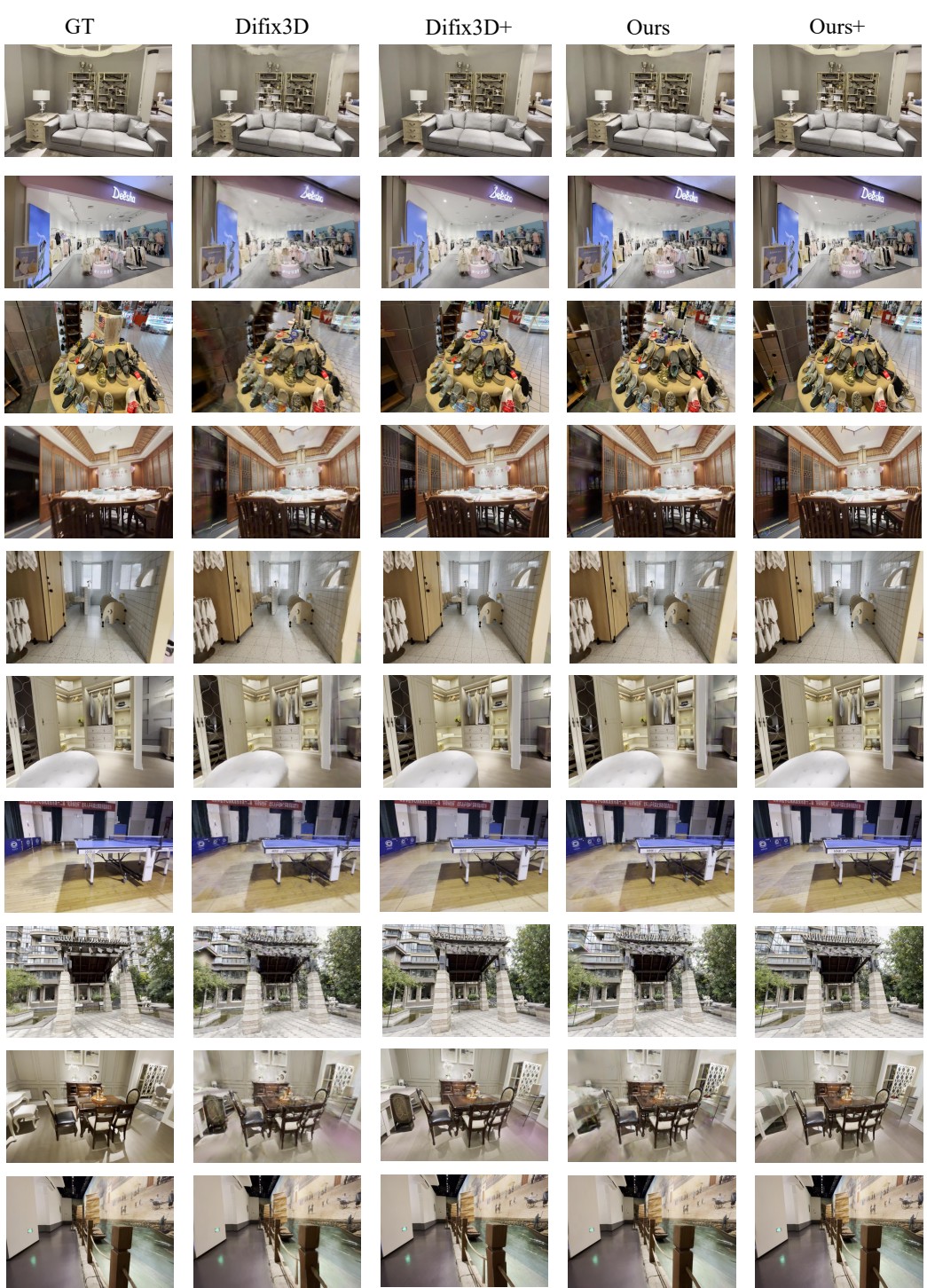

Figure 7: **More visual comparisons on the DL3DV-10K dataset (Ling et al. (2024)).**

