# OpenReview forum: "FixingGS: Enhancing 3D Gaussian Splatting via Training-Free Score Distillation"
_ICLR.cc/2026/Conference — ICLR 2026 Conference Withdrawn Submission_

### Official Review · Reviewer_4Fn1 · 2025-10-25

**Soundness:** 2
**Presentation:** 3
**Contribution:** 2
**Rating:** 2
**Confidence:** 4

**Summary:**

This work proposes a framework FixingGS for enhancing the sparse-view 3DGS, which includes the training-free score distillation of the diffusion model, and the adaptive progressive enhancement strategy for the guidance of the diffusion prior. Experiments on different datasets show FixingGS achieves improvements compared to several baselines.

**Strengths:**

* This paper is well-written, and the readers can easily understand the points authors want to present.
* The proposed adaptive progressive enhancement strategy seems intresting for me.

**Weaknesses:**

* One of the main part of this work "training-free score distillation" is not novel enough, and I think it is a quite normal technique for extracting the diffusion knowledge.
* Although the proposed adaptive progressive enhancement strategy seems intresting, it's more of an engineering skill than a fundamental innovation.
* The experimental results are hard to prove the effectiveness of the proposed method compared to other methods, e.g., FixingGS only achieves the improvement of +0.003 SSIM on the DL3DV-10K dataset compared to FSGS, which does not utilize the external diffusion models, but utilizes the monocular depth prediction model as its main external supervision.
* The ablations are also hard to prove the effectiveness of the proposed FixingGS. The experimental results in Table 3 show limited improvements of the proposed modules.

**Questions:**

There are no special questions. See weaknesses for further discussions.

---

### Official Review · Reviewer_q4vs · 2025-10-25

**Soundness:** 2
**Presentation:** 3
**Contribution:** 2
**Rating:** 4
**Confidence:** 4

**Summary:**

The paper targets sparse-view novel view synthesis under challenging cases. It proposes an iterative loop: render → diffusion repair → reliability gating via a PSNR threshold → adaptive pose shifting toward better-constrained regions → feed repaired views back into training. The idea is practically motivated, and the pipeline is clearly written. Reported results show improvements over several baselines on standard image-quality metrics.

**Strengths:**

1. **Practical motivation & clear pipeline.** Using diffusion as a prior to “repair” sparse-view renderings and feeding the signal back into 3DGS is a sensible, implementable loop.
2. **Handling sparse and blurred scenes.** The paper targets areas where sparse view reconstruction is particularly prone to failure. This is a real problem with obvious practical pain points, not just a small indicator stacking under normal conditions.
3. **Clarity-wise:** the manuscript is clearly written, with well-structured methodology, detailed explanations, and intuitive visualizations that enhance understanding

**Weaknesses:**

**1. Hard-coded PSNR threshold (η) lacks justification, parameter sweep, or visual evidence**
The gating rule “trigger APE if PSNR(I_fix, I_extra) < η” is central to the proposed safety mechanism. However, the rationale for using PSNR instead of alternative similarity measures such as SSIM, LPIPS, or a composite metric is not explained. The method for selecting η (fixed or adaptive) is also unclear. The paper provides no histogram of PSNR values, no ablation over η (for example, 18, 20, 22, 24, 26, or 28 dB), and no visual examples demonstrating why low PSNR corresponds to unreliable outputs.

**2. Supervision configuration and hyperparameter ablations are insufficient**
The study fixes a single “pseudo-label alignment” configuration, where diffusion-repaired images are used as targets under a pixel-level loss. Several key hyperparameters, including t₀, the global weighting term ω, and optional perceptual terms, are fixed without robustness analysis. This limits the understanding of how sensitive the approach is to supervision choices.

**3. Consistency metrics and failure analysis are limited**
The paper repeatedly claims that the proposed method improves cross-view consistency and mitigates multi-view inconsistency. Table 3 presents an ablation comparing “with APE” and “without APE” using standard reconstruction metrics (PSNR, SSIM, and LPIPS), which demonstrates overall quality gains but does not substantiate the claimed mechanism. Appendix G introduces a cross-view consistency metric (for example, TSED) evaluated on DL3DV, which is informative but has two limitations: (i) it is not included among the main results or applied to other benchmarks, and (ii) it is not linked to mechanism-specific ablations such as with versus without APE under TSED, or under different reliability thresholds η. Moreover, only successful cases are shown, without any examples of failure modes such as diffusion hallucinating textures that fail to remain consistent across nearby views.

**4. Lacking Training Details**
Diffusion-based backbones typically require input resolutions aligned with network strides (for example, multiples of 8). The paper does not specify the native resolutions used for DL3DV or other datasets, nor does it describe the preprocessing strategy applied to satisfy this constraint. It is also unclear whether isotropic resizing, center cropping, or reflection padding was used, and whether the same pipeline was applied to all baselines. These choices can substantially affect image sharpness, field of view, and fairness across comparisons.

---
I have listed my concerns, and the score will be adjusted based on the author's response.

**Questions:**

Please refer to Weaknesses part.

---

### Official Review · Reviewer_fGZm · 2025-10-27

**Soundness:** 2
**Presentation:** 3
**Contribution:** 2
**Rating:** 4
**Confidence:** 4

**Summary:**

This paper aim to address the artifact issues in 3D Gaussian Splatting (3DGS) when reconstructing from sparse viewpoints. Recent methods that enhance 3DGS with generative priors suffer from the problem of using "lagging" diffusion priors that are updated at fixed intervals, leading to inconsistent supervision. The proposed method, FixingGS, introduces a "training-free" score distillation approach that continuously leverages a pre-trained diffusion model (Difix) throughout the optimization process to provide more accurate and cross-view consistent guidance. Additionally, an Adaptive Progressive Enhancement (APE) scheme is proposed to improve under-constrained regions by identifying unreliable views and generating new training samples using multiple reference views.
Experiments on the DL3DV-10K and Mip-NeRF 360 datasets demonstrate superior quantitative and qualitative results compared to existing SOTA methods.

**Strengths:**

- The proposed method is easy to implement and follow.
- The proposed method APE shows good performance in the experiments, achieving better results than existing SOTA methods on the DL3DV-10K and Mip-NeRF 360 datasets, the ablation studies demonstrate the effectiveness of each component of the method.

**Weaknesses:**

- This paper suffers from lack of novelty. The proposed method does not make significant improvements over Diffx3D+, mainly replacing the "lagging" diffusion prior in DiffGS with a "training-free" score distillation approach, without any improvements on SDS itself. There is no modification during the distillation process. The claimed score distillation introduces higher computational overhead during training (as it requires calling the Diffx3D+ model at every training step), while the choice of Diffx3D+ seems to be merely for efficiency balance.
- The experimental settings are not reasonable. The comparison is made with the official Diffx3D+ results, but if the claim is that the "lagging" diffusion prior in Diffx3D+ is problematic, why not simply reduce the update interval of Diffx3D+ to make it update the diffusion prior more frequently? This could also address the "lagging" issue. The paper does not provide relevant comparative experiments. Additionally, the paper should compare with more methods that use SDS.
- The APE method uses a heuristic approach to select unreliable views, which is relatively crude. The authors could consider improving it by referencing adaptive methods like FisherRF

[1] Active View Selection and Mapping with Radiance Fields using Fisher Information

**Questions:**

- Why not reduce the update interval of Diffx3D+ to make it update the diffusion prior more frequently, instead of introducing a new score distillation method? Have you conducted any experiments to compare this approach?

---

### Official Review · Reviewer_E9KJ · 2025-10-28

**Soundness:** 3
**Presentation:** 3
**Contribution:** 3
**Rating:** 4
**Confidence:** 5

**Summary:**

This paper integrates the SDS loss into 3D reconstruction with diffusion priors, achieving superior results compared to using diffusion-refined images directly. To address unreliable views, it further introduces the APE module, which leverages multiple reference views with slightly shifted camera poses to enhance reconstruction consistency. Experimental results show that APE effectively improves the overall reconstruction quality.

**Strengths:**

1. This work is the first to apply the SDS loss for novel view synthesis, demonstrating state-of-the-art performance. The results clearly show that leveraging diffusion gradients provides a more effective supervision signal than directly using diffusion-refined images, highlighting a significant novelty and contribution in bridging diffusion priors with 3D reconstruction.

2.The proposed APE module effectively enhances the performance of the diffusion + 3DGS pipeline, particularly under unreliable or challenging view conditions.

3.The paper is well written, logically structured, and easy to follow.

**Weaknesses:**

1. Based on Equation (5), the essential difference between the proposed SDS loss and DiFix3D lies in the additional gradient path that passes through the diffusion model. This extra gradient allows 3DGS to receive guidance not only from the pixel-space reconstruction loss but also from the diffusion model’s internal score field, which theoretically provides distribution-level supervision. However, the paper does not analyze why or how this additional gradient benefits the 3D reconstruction process. For instance, it remains unclear whether the score-based gradient improves geometry consistency, enhances perceptual realism, or simply acts as a regularization term.

2. The novelty of this work appears somewhat incremental. Integrating the SDS loss into a 3DGS pipeline has already been explored in prior works such as DreamGaussian [1] . Moreover, it is well known that SDS loss often introduces blurring artifacts and degrades high-frequency fidelity, which was well studied in VSD[2]. The authors should make their framework more solid—e.g., by analyzing how their pipeline mitigates the typical limitations of SDS or by demonstrating clear advantages beyond a simple replacement of the loss function.

3.Using SDS loss alone does not automatically make the pipeline training-free. The framework remains training-free only if the diffusion prior is a frozen, general pretrained model. Once a task-specific or fine-tuned diffusion (e.g., DiFix3D+) is used, the pipeline effectively relies on prior training and thus cannot be strictly regarded as training-free.

4.As stated in the paper, the SDS loss is applied throughout the entire training process, which is expected to substantially increase the overall training time due to repeated diffusion inference and backpropagation through the score network.

Here is the citation:
[1] Tang, J., Ren, J., Zhou, H., Liu, Z., & Zeng, G. (2023). Dreamgaussian: Generative gaussian splatting for efficient 3d content creation. arXiv preprint arXiv:2309.16653.
[2] Wang, Z., Lu, C., Wang, Y., Bao, F., Li, C., Su, H., & Zhu, J. (2023). Prolificdreamer: High-fidelity and diverse text-to-3d generation with variational score distillation. Advances in neural information processing systems, 36, 8406-8441.

**Questions:**

1. Does the SDS loss make the scene more blurry?

2 Why does 3DGS outperform many methods on the DL3DV dataset?

3.Since the SDS loss is applied throughout the entire training process, have you tried doing the same for DiFix3D+?

---

### Note · Authors · 2025-11-19

I have read and agree with the venue's withdrawal policy on behalf of myself and my co-authors.